# Efficient Lead Pb(II) Removal with Chemically Modified *Nostoc commune* Biomass

**DOI:** 10.3390/molecules28010268

**Published:** 2022-12-28

**Authors:** Carmencita Lavado-Meza, Leonel De la Cruz-Cerrón, Carmen Lavado-Puente, Julio Angeles-Suazo, Juan Z. Dávalos-Prado

**Affiliations:** 1Facultad de Ingeniería, Universidad Continental, Huancayo 12003, Peru; 2Escuela Profesional de Ingeniería Ambiental, Universidad Nacional Intercultural de la Selva Central, Chanchamayo 12856, Peru; 3Facultad de Ingeniería Industrial, Universidad Tecnológica del Perú, Lima 15046, Peru; 4Instituto de Química Física “Rocasolano”, CSIC, 28006 Madrid, Spain

**Keywords:** biosorption, Pb(II) removal, *Nostoc commune*, cyanobacteria

## Abstract

A new biosorbent based on *Nostoc commune* (NC) cyanobacteria, chemically modified with NaOH (NCM), has been prepared, characterized and tested as an effective biomass to remove Pb(II) in aqueous media. The adsorption capacity of NCM was determined to be q_e_ = 384.6 mg g^−1^. It is higher than several other biosorbents reported in the literature. Structural and morphological characterization were performed by FTIR, SEM/EDX and point zero of charge pH (pH_PZC_) measurements. NCM biosorbent showed more porous surfaces than those NC with heterogeneous plates including functional adsorption groups such as OH, C = O, COO^−^, COH or NH. Optimal Pb(II) adsorption occurred at pH 4.5 and 5.5 with a biomass dose of 0.5 g L^−1^. The experimental data of the adsorption process were well fitted with the Freundlich-isotherm model and pseudo-2nd order kinetics, which indicated that Pb(II) adsorption was a chemisorption process on heterogeneous surfaces of NCM. According to the thermodynamic parameters, this process was exothermic (∆*H*^0^ < 0), feasible and spontaneous (∆*G*^0^ < 0). NCM can be regenerated and efficiently reused up to 4 times (%D > 92%). NCM was also tested to remove Pb (%R~98%) and Ca (%R~64%) from real wastewater.

## 1. Introduction

Nowadays, worldwide, there has been an increase in the demand for drinking water. However, there is general neglect regarding the control and prevention of various forms of contamination that affect the potential sources of water supply and storage. One of the contaminants of concern are heavy metals, due to their toxic, persistent and bioaccumulative nature, which seriously affect human health and the environment [1]. This is the case of Pb(II), which accumulates in the bones (half-life of 20 years), affects the nervous [2] and reproductive systems, red blood cells and kidneys [3], in addition to being a potential carcinogen, given its high affinity for the thio (–SH), oxo (–O) and phosphate (PO_4_^3−^) groups of some enzymes, ligands and biomolecules.

The main removal techniques for heavy metals include methods such as coagulation, chemical precipitation, electrodialysis, evaporative recovery, flotation, flocculation, ion exchange, nanofiltration, reverse osmosis, ultrafiltration, etc. [4]. Although these methods are effective, they are not economical due to the high costs in energy and production reagents. These can also generate considerable amounts of sludge and toxic by-products. Therefore, it is important to develop efficient, economical and, above all, environmentally friendly methods, capable of effectively removing toxic products [5,6,7]. Within this category are the so-called biosorbents, which by biosorption processes eliminate metals and other toxic products from an aqueous solution. Biosorption includes absorption, adsorption, ion exchange, surface complexation and precipitation processes.

Biosorbents can be microbial, such as bacteria (e.g., cyanobacteria), fungi, yeasts, algae [8,9] and also plant materials and agro-industrial waste [10], that due to the presence on their surfaces of functional groups, such as hydroxyls, carbonyls, phosphates, amines or thiols, can efficiently remove a wide variety of metallic ions [11].

Cyanobacteria are microorganisms that inhabit a wide variety of environments, perform photosynthesis and form a thick microbial layer even in extreme habitat settings [12]. Used as biosorbents, they have interesting properties, such as a large surface area and a larger volume of mucilage with high binding affinity and simple nutrient requirements [13]. These microorganisms can produce considerable amounts of extracellular polymeric compounds including proteins, exopolysaccharides, uronic acids and humic and lipid compounds [14]. Among these components, extracellular exopolysaccharides stand out as powerful chelating agents capable of trapping metallic ions in solution [5,15].

*Nostoc commune* (NC), or blue-green algae, is a cyanobacterium that has gelatinous matter on its surface, contain little protein but a lot of dietary fiber with minerals such as Ca or Fe [16] and reproduces in large numbers on ponds, ditches, waterlogged places and nutrient-poor limestone soils [17]. NC is widely distributed throughout the world, from polar to tropical regions, making it an abundant and cheap biomass. Previous studies revealed the effectiveness of NC for the elimination of several contaminants in wastewater [18]. Proper treatment prevents its leaching and increases the concentration of active sites, which leads to an improvement in its metallic biosorption capacity. Treatments with acids (HNO_3_, HCl, H_3_PO_4_, citric acid, etc.) and bases (NaOH, KOH) have been reported, the latter being the ones that gave the best biosorption results [19,20,21]. In this work we report the removal capacity of Pb(II), in aqueous solutions, of the *Nostoc commune* biomass, chemically modified with NaOH and coming from Mantaro valley of the Junin Region (Peru).

## 2. Results and Discussion 

### 2.1. Effect of Alkaline Treatment, Concentration of Acidic and Basic Sites, pH_PZC_ Determination

Table 1 shows the concentration of acidic and basic sites on the NC (untreated) and NCM (treated) biomasses. The alkaline treatment of NC is reflected in a significant increase in the concentration of basic sites (almost six times) and a decrease in acid sites on the NCM sample. According to Bulgariu et al. [22,23], the increase in basic sites in biomasses—such as *Ulva lactuca* algae—treated with NaOH would be due to hydrolysis reactions that would increase, on the cell surfaces, the formation of carboxylic (–COO^−^) and hydroxyl (–OH) groups. The presence of these groups on NCM has been confirmed by our FTIR analyses.

The point zero of charge pH values (pH_PZC_) were deduced from the ΔpH (=pH_0_ − pH_f_) vs. pH_0_ plot (See Figure 1). The two curves belonging to the NC and NCM intersect the pH_0_–axis at pH_PZC_ = 1.3 and 2.5, respectively. It is important to mention that pH_PZC_ provides valuable information on the electrostatic interactions between sorbents and sorbed species, thus for pH < pH_PZC_, the sorbent–surface charge is positive, therefore it can interact with anionic sorbates. On the contrary, at pH > pH_PZC_ the sorbent–surface charge is negative and it can interact more with cationic sorbates. As can be seen in Figure 2, the pH_PZC_ for NCM (2.5) is higher than for NC (1.3). This result indicates an increase in the superficial alkalinity of NCM compared to NC, which is consistent with the increase in the concentration of its basic sites, described above. In this regard, Šoštarić et al. [11] reported similar results, with the alkaline treatment of the apricot shell biomass.

An interesting consequence of the alkaline treatment of NC that increases adsorption basic sites and pH_PZC_ value is the significant increase in its Pb(II) removal capacity, q_e_. Taking into account optimal conditions, described below, the q_e_ of NCM is 1.6 times higher than for untreated biomass NC (See Figure 2).

### 2.2. SEM/EDX Morphological and Structural Characterization, FTIR Analysis

Figure 3 shows the morphologies of NC, NCM and the last one loaded with Pb(II) (NCM-Pb). NCM shows a more porous and cracked surface than NC. According to Zafar et al. [24], the treatment with NaOH would eliminate the protein and lipid fractions of the biomass, generating a greater contact area and thus increasing the adsorption capacity of Pb(II) from the treated surface. Iddou et al. [19] reported similar results, showing more porous surfaces on brown algae treated with NaOH, with respect to the precursor biomass. Pb(II) adsorption changes the morphology of NCM, making the surface less rough and porous (Figure 3c). 

Figure 3 (right) shows the EDX spectra of NC, NCM and NMC-Pb. We can appreciate the presence of common peaks associated with C, O, N, Al and Si. NC and NCM also show the presence of Mg, and only in NCM is Na present. In NCM-Pb, the presence of Pb peaks is notable, as is the absence of Na and Mg peaks. El-Naggar et al. [25] reported similar results after loading with Pb(II) the biomass of the marine algae *Gelidium amansii*. The disappearance of the Mg and Na peaks on the surface of NCM-Pb (Figure 4c) would be due to the interaction of ion exchange and/or complexation of Pb(II) with monovalent (H^+^, Na^+^) or divalent (Mg^2+^) cations of the biomass, which would be released by Pb(II) [3,26].

#### FTIR Analysis

FTIR spectra (Figure 4 left) of the NCM show band positions at:

1/3427.3 cm^−1^ associated with the OH groups of polymeric compounds (such as sucrose) and the NH groups of proteins [27].

2/2921 cm^−1^ attributable to the symmetric stretching of the aliphatic chains of C–H bonds of alkyl functional groups in carbohydrates [27].

3/1639.4 cm^−1^ associated with the stretching of carbonyl group bonds (C=O), primary and secondary amides of protein peptide bonds [9].

4/1537.2 cm^−1^ associated with carboxylates (COO^−^) [15,28].

5/1417.6 and 1035.5 cm^−1^ would be due to the stretching of C–H and C–OH bonds respectively in pyranose units [29,30].

FTIR spectra of NCM loaded with Pb(II), NCM-Pb (Figure 4 right) show changes, with respect to NCM, in the intensity and position of some adsorption peaks. Thus, the positions of the 3/, 4/ and 6/ bands are significantly displaced at Δ_3_ = 4.0, Δ_4_ = 9.6, Δ_6CH_ = 3.8 and Δ_6COH_ = −9.4 cm^−1^. The displacement of these bands proves the participation of carbonyl, amide and carboxylic groups in the Pb(II) adsorption process [31].

### 2.3. Influence of pH Solution

pH is an important factor in the biosorption of heavy metals, given that the speciation and removal of heavy metals in aqueous solution is highly dependent on this factor. According to Joseph et al. [32], there is a consensus that low pH values (<4) hinder the adsorption of heavy metals, while pH values between 5 and 7 are the most effective. In this context, we have studied the effect of pH on the Pb(II) absorption capacity q_e_, at different C_0_ (initial concentration of Pb ions) values. The pH range considered was 2.5 to only 5.5 because Pb precipitates at higher values, forming basically Pb(OH)_2_ (See Figure 5, speciation of Pb). We can observe that the trend of each q_e_ vs. pH (Figure 5) curve is similar for all considered C_0_ values, that is, q_e_ increases with increasing pH until reaching a plateau between pH 4.5 and 5.5. It is interesting to note that, even at low pHs, the q_e_ values for NCM are significant. This result would be due to NCM having a pH_PZC_ = 2.5, which indicates the basic nature of its surface, with numerous active sites, such as –OH, –COOH, –COH and –NH functional groups capable of favoring the adsorption of Pb(II). The increments of biosorption until pH 5.5 could be associated also with the deprotonation of carboxyl, hydroxyl and other negatively charged groups, leading to the electrostatic attraction of positively charged Pb(II) [4,33].

### 2.4. Influence of NCM Dose and Initial Concentration of Pb(II) Ions, C_0_

Figure 6 shows the influence of the biomass dose of NCM on the adsorption capacity q_e_ and on the %R removal percentage of Pb(II). For each initial concentration C_0_, the same trend is observed in both q_e_ vs. dose (descending) and in %R vs. dose (ascending) plots, that is: i/significant increase in %R up to an NCM dose of 0.5 g L^−1^, from which this increase is less pronounced, particularly for the lowest C_0_ (127.3 mg L^−1^) where %R reaches almost 97%. On the contrary, for the highest C_0_ (311.1 mg·L^−1^), %R reaches almost 64%, which would be related to the decrease and saturation of active adsorbent sites due to the agglomeration of biomass particles [34]. ii/For a given C_0_, q_e_ decreases with increasing NCM dose. This behavior was also reported by Mangwandi et al. [35], in the removal of Cr(IV) with date pits and olive stones.

Taking in account these results, we consider 0.5 g·L^−1^ the optimal NCM dose for Pb(II) removal.

### 2.5. Kinetic of Biosorption 

We can note (Table 2) a better correlation (R^2^ = 1) with *pseudo*-second order than with the first-order adjustment models, although this last one, according to Vijayaraghavan et al. [36], can be applied during the initial periods of the biosorption process. Accordingly, we can affirm that i/the adsorption of Pb(II) is a chemisorption process wherein the sorption capacity is related to the number of occupied active sites [37]; ii/the calculated biosorption capacity at time t, q_t_, is very close to the corresponding experimental value; iii/the calculated biosorption capacity at equilibrium, q_e,cal_ = 384.6 mg g^−1^, is the value that we consider the maximum q_e_ value (q_e,max_) for the Pb(II) removal capacity of NCM. 

The *pseudo*-second order kinetic for NCM is the same model that governs the Pb(II) adsorption kinetics of several biosorbents, such as taro [38], olive pit [4], *Geldium amansii* [25] or shells of peanut [39].

There is an acceptable correlation (R^2^ = 0.8) for the Elovich kinetic model, which assumes that the active sites of the sorbent are heterogeneous and therefore exhibit different sorption energies. The α rate constant value is comparable to that obtained by Elwakeel et al. [40], which can be attributed to the negative surface-charge nature of the sorbent that allows the rapid sorption of Pb(II).

The q_t_ vs. t^0.5^ plot (Figure 7) was fitted to the intra-particle diffusion Weber–Morris model. We can distinguish three parts: the first part, showing a rapid growth of q_t_ at the time t (0 < t^1/2^ < 3) which would indicate the rapid absorption of Pb(II) ions on the outer surface of the biosorbent; the second part, a slower growth of q_t_ with t (3 < t^1/2^ < 5.5), which would be related to a gradual adsorption process, wherein Pb ions would enter and fill the biosorbent pores and the intraparticle diffusion would be the rate-limiting step [41,42]; finally, the third part (t^1/2^ > 5.5), wherein q_t_ is practically constant (plateau region), would indicate the equilibrium uptake, wherein the intra particle diffusion is not the only rate-controlling step [34].

### 2.6. Adsorption Isotherms

Data obtained from Pb(II) biosorption experiments were fitted to Langmuir, Freundlich and Dubinin–Radushkevich (D–R) equilibrium models. The isotherms were determined for optimal conditions at pH 4.5 and 5.5, room temperature, NMC dose 0.5 g L^−1^, contact time t = 60 min and a range of C_0_ concentrations between 127 and 442 mg L^−1^. Figure 8 shows adsorption isotherms fitted to Langmuir and Freundlich models.

The determined parameters are consigned in Table 3. We can see that the experimental data of the adsorption isotherms fit well to the Langmuir model, particularly at pH 4.5, with parameters q_max_ (maximum adsorption capacity) and K_L_ practically similar to those obtained at pH 5.5. It is interesting to mention that the value of q_max_ = 384.6 mg g^−1^ is the same as the q_e,max_ calculated by the *pseudo*-second order kinetic.

Nevertheless, a best fit, at pH 4.5, is obtained with the Freundlich model, given the lowest χ^2^ value with R^2^ close to 1. The results indicate that the biosorption of Pb(II) on NCM is a chemisorption process (n_F_ >1) with a high affinity between Pb ions and the heterogeneous surfaces of NCM [43].

The fits of the biosorption data to the Dubinin–Radushkevich (D–R) model are acceptable. The values of q_max_ obtained are close to those determined with the Langmuir model (deviation between 15 and 20%), and the values of the parameter E (>500 kJ mol^−1^) confirm that the biosorption of Pb(II) on NCM is a chemisorption process [41].

The fits to the Temkin model give low χ^2^ values with R^2^ > 0.9. The parameter B (>75 J mol^−1^) values, related to the heat of biosorption [44,45], reinforce the chemical–biosorption mechanisms already indicated by the Freundlich and Dubinin–Radushkevich models.

The maximum Pb(II) biosorption capacity q_e,max_ values of biomasses and agro-industrial wastes similar to that studied in this work are shown in the Table 4. It is interesting to note that the q_e,max_ of NCM is among the highest.

### 2.7. Biosorption Thermodynamics

For the thermodynamic study, the Van’t Hoff equation (1) was used, which is derived from the equation $\Delta G^{o}=\Delta H^{o}-T\Delta S^{o}$ that relates the Gibbs energy (Δ*G*^0^), enthalpy (Δ*H*^0^) and entropy (Δ*S*^0^) of the adsorption process. Δ*G*^0^, at different temperatures, were evaluated from the expression $\Delta G^{0}=-RT\ln\left(K_{c}\right)$ where *K_c_* is the equilibrium constant, and $K_{c}=\frac{C_{\mathrm{es}}}{C_{e}}$; C_es_ and C_e_ are the equilibrium Pb(II) concentrations, respectively, in the biosorbent and in the solution. R is the universal gas constant, and T is the temperature of the solution.
(1)$$\ln\;K_{c}=\frac{\Delta S^{0}}{R}-\frac{\Delta H^{0}}{RT}$$

The parameters obtained are consigned in Table 5. The negative values of Δ*G*^0^ indicate that the Pb(II) adsorption process on NCM is spontaneous and favorable for all the considered temperatures (see Table 5). The negative value of the entropy change Δ*S*^0^ proves the decrease in randomness at the solid/solution interface during biosorption. On the other hand, the negative value of Δ*H*^0^ indicates that the process studied is exothermic, and its magnitude (65.1 kJ mol^−1^), according to Cardoso et al. [49], confirms the chemical nature of the Pb(II) adsorption process (chemisorption). Similar results were found in the removal of Pb(II) with *Cystoseira stricta* marine algae [19] and also with raw/treated maize stover [46]. 

### 2.8. NCM for the Pb(II) Removal in Real-Wastewater

For the possible application of NCM in the treatment of contaminated water, we consider real wastewater that contains, in addition to Pb, other metals such as Ca, K and Na, whose concentrations are shown in Table 6. For the removal of Pb(II), it has been considered optimal biosorption conditions, which were determined in this work: dose of NCM = 0.5 g L^−1^ and pH 5.5.

We can appreciate significant percentages of removal (%R) of Pb(II) (~98%) and Ca (~64%), to a lesser extent of K (10%) and to a practically null amount of Na (Table 6).

The results obtained show that NCM is a cheap, abundant and very effective biomass to remove Pb(II) and is also effective to remove Ca from real wastewater with high concentrations of these metals.

### 2.9. Regeneration of NCM Biosorbent

The adsorption/desorption studies are related to the economic and environmental viability of the sorbent material, since good stability (effective regeneration) and multiple re-uses of it are desirable [50]. We can see in Figure 9, that after four adsorption/desorption cycles, the regeneration efficiency, %D, of NCM is greater than 92%. This result shows that NCM is a stable, reusable and efficient biosorbent for the aqueous removal of Pb(II). In this regard, Bangaraiah et al. [50] removed Pb(II) with chemically modified green algae, reusing it up to three times with %D = 87%.

## 3. Materials and Methods

### 3.1. Preparation of Nostoc Commune Biosorbent

Untreated biomass (NC)

Untreated biomass was obtained from gelatinous colonies of *Nostoc commune* (Figure 10) which were collected at Pampa Cruz in the Junín Region of Peru. The material was washed, then dried in an oven at 333 K for 48 h and finally ground.

Treated biomass (NCM)

NC was placed in contact with 0.1 M NaOH solution in a ratio of 1 g:10 mL for 24 h with constant stirring at 300 rpm. Afterwards, the mixture was filtered and washed with abundant deionized water until reaching a colorless solution and neutral pH. Finally, the sample was dried at 333 K for 48 h, then it was ground and sieved with a 0.3 mm mesh. 

### 3.2. Biosorbent Characterization

The point zero of charge pH values (pH_PZC_) were determined according to the procedures described by do Nascimento et al. [20]. It has been prepared as a mixture of 0.05 g of biosorbent with 50 mL of aqueous solutions under different initial pHs (pH_0_) ranging from 1 to 8. The acid solutions were prepared from 1 M HCl, while the basic solutions were prepared from 1 M NaOH. After 24 h of equilibrium, the final pHs (pH_f_) were determined.A Fourier transform infrared spectrophotometer (FTIR, SHIMADZU- 8700) was used to identify the functional groups present on the surface of biosorbents. The wavelength was set to 4000 to 400 cm^−1^.Morphological and elemental analysis of the biosorbent surface were performed by scanning electron microscopy (SEM) coupled with EDX (energy-dispersive X-ray spectroscopy) (Hitachi SU8230 model).The concentration of acid and basic sites was determined by the Boehm method following the procedures described by do Nascimento et al. [51].

All measurements were performed in triplicate.

### 3.3. Biosorption Assays

Experimental parameters such as pH, biosorbent dosage, initial and final or equilibrium Pb(II) concentrations and those related to kinetic and isothermal properties of Pb(II) biosorption on NC and NCM samples have been determined. Between 0.0125 and 0.5 g of each biomass was added to 50 mL of Pb(NO_3_)_2_ solution with a Pb(II) initial concentration between 127.3 to 440.4 mg L^−1^. These solutions had been adjusted to a pH in the range of 2.5 to 5.5 by adding 0.1 M HNO_3_ or 0.1 M NaOH. The suspension obtained was stirred at 300 rpm for a time period of 60 min. The temperature was kept at 293 K (room temperature). The Pb(II) adsorption capacity, q*_e_* (in mg·g^−1^), onto biosorbents was determined by Equation (2) measuring the concentration of Pb before and after the biosorption process by means an atomic absorption spectrophotometer (SHIMADZU-AAS 6800). The Pb(II) removal efficiency (%R) was calculated by using Equation (3)
(2)$$q_{e}=\frac{C_{0}-C_{e}}{m}\times V$$
(3)$$\%R=\frac{C_{0}-C_{e}}{C_{0}}\times 100$$
where C_0_ and C_e_ (in mg L^−1^) are the initial and equilibrium or final Pb(II) concentrations, respectively; V(in L) is the volume of solution and m (in g) is the biosorbent mass. The adsorption experiments were repeated three times, and the average values were reported.

The experimental data of isotherms and biosorption kinetics on NC and NCM were evaluated and correlated with different models, described in Table 7. The quality of the corresponding adjustments is reflected by chi-squared χ^2^ and/or correlation coefficient R^2^ values [52].

### 3.4. Desorption Experiments

A total of 50 mg of NCM biosorbent previously loaded with Pb(II), from the mixture with 100 mL of Pb solution ([Pb(II)] = 127.3 mg L^−1^), was then then filtered, washed and dried, then was subjected to the desorption process by adding 50 mL of 0.1 M HNO_3_ eluent and then stirring the result at 200 rpm for 60 min. After that, the biosorbent was washed with distilled water, dried and re-used again. The adsorption/desorption operation was repeated up to four times. The [Pb(II)] concentration adsorbed and desorbed was analyzed by an atomic absorption spectrophotometer (before being described).

The desorption or regeneration efficiency (%*D*) of the NCM biomass was calculated using the following expression [53]: (4)$$\%D=\frac{Pb\left(II\right)desorbed\;from\;biomass}{Pb\left(II\right)adsorbed\;on\;biomass\;}\times 100$$

## 4. Conclusions

*Nostoc commune* cyanobacteria was chemically modified, with a 0.1 M NaOH. The Pb(II) adsorption capacity of the treated biomass (NCM) was almost 1.6 times higher than for untreated biomass (NC).Point zero of charge, pH_PZC_, of NCM (= 2.5) was greater than for NC (=1.3). It is consistent with the concentration of basic sites, which is almost six times higher for treated than untreated biosorbents. The basic sites would be associated with OH, C=O, COH, COO^−^ and NH functional groups (identified by FTIR).SEM/EDX analyses of NCM showed a more porous and cracked surface than NC, but once charged with Pb(II), the morphology changed to a less rough and porous surface.For a given initial Pb ions concentration, C_0_, the biosorption capacity q_e_ of NCM reached a maximum plateau at a pH between pH 4.5 and 5.5.We consider 0.5 g L^−1^ the optimal NCM dose for Pb(II) removal, given that the corresponding efficiency, %R, can reach almost 97% for low C_0_ values.The adsorption kinetic data were well fitted with the *pseudo*-second order model, indicating that the Pb(II) biosorption on NCM was a chemisorption process, with a removal capacity of q_e_ = 384.6 mg g^−1^. The Elovich kinetic model indicated a rapid sorption of Pb(II). It is consistent for the first stage described by the intra-particle diffusion Weber–Morris model, since, in the following stages, Pb(II) adsorption was very slow.The adsorption isotherms’ data were well fitted with the Freundlich model (heterogeneous adsorption) at pH 4.5 and 5.5. Freundlich, Dubinin–Radushkevich and Temkin models confirmed that Pb(II) adsorption on NCM is a chemisorption process, which was thermodynamically characterized as exothermic (Δ*H^0^* < 0), feasible and spontaneous (Δ*G^0^* < 0) and with a decreasing randomness (Δ*S^0^* < 0) at the solid/liquid interface.The maximum Pb(II) biosorption capacity of NCM, q_e,max_= 384.6 mg g^−1^, is higher than for other similar treated biosorbents reported in the literature.Desorption–regeneration experiments showed that NCM can be recovered efficiently (%D > 92%) up to four times.NCM was tested as an inexpensive and efficient biosorbent to remove Pb and Ca, from real wastewater, with an efficiency %R of almost 98% and 64%, respectively.

## Figures and Tables

**Figure 1 molecules-28-00268-f001:**
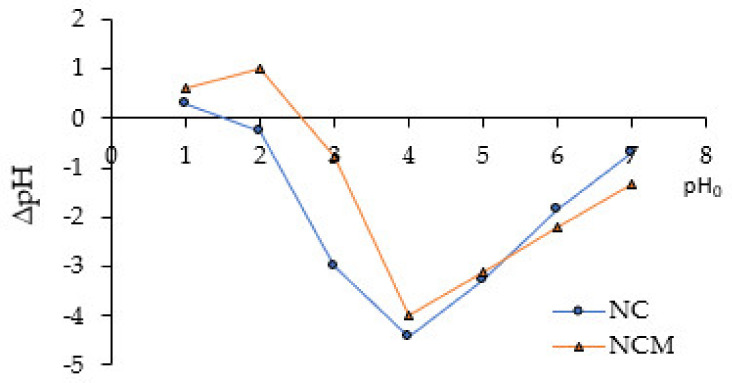
Determination of pH_PZC_ values for NC and NCM biosorbents.

**Figure 2 molecules-28-00268-f002:**
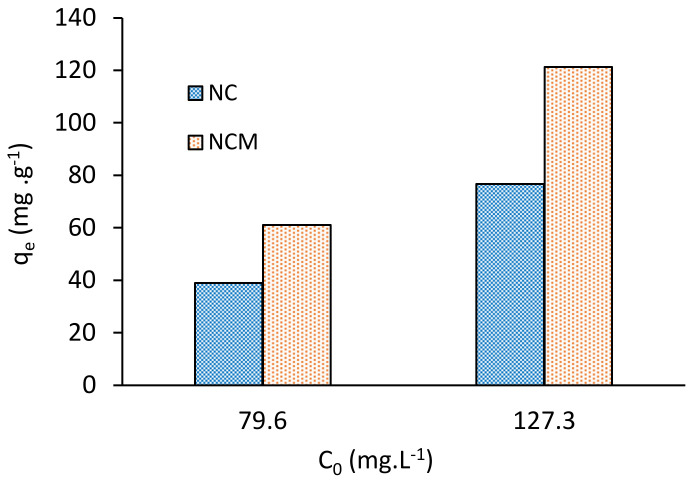
Pb(II) adsorption capacities, q_e_, of treated (with NaOH) NCM and untreated NC biomass. pH 4.5, t = 60 min, biomass dose = 0.5 g L^−1^, T = 293 K.

**Figure 3 molecules-28-00268-f003:**
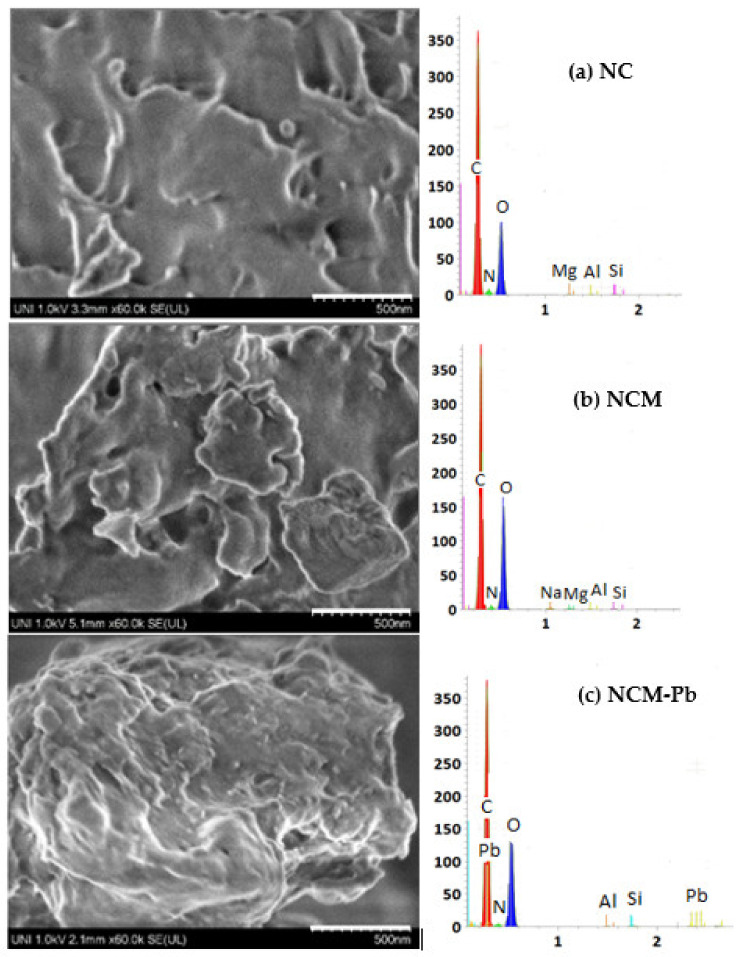
SEM images (**left**) and EDX spectra (**rigth**) of NC (**a**), NCM (**b**) and NCM-Pb (**c**).

**Figure 4 molecules-28-00268-f004:**
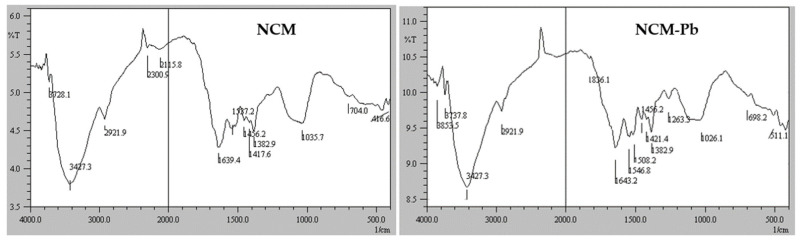
FTIR spectra of the NCM before (**left**) and after Pb(II) biosorption (**right**).

**Figure 5 molecules-28-00268-f005:**
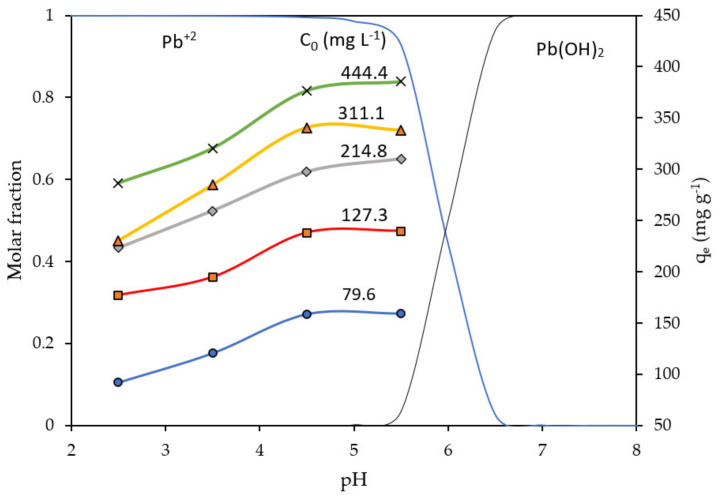
q_e_ and Pb(II) species distribution vs. pH plots at different C_0_ initial concentrations of Pb ions. T = 293 K, t = 60 min, biosorbent dose = 0.5 g L^−1^.

**Figure 6 molecules-28-00268-f006:**
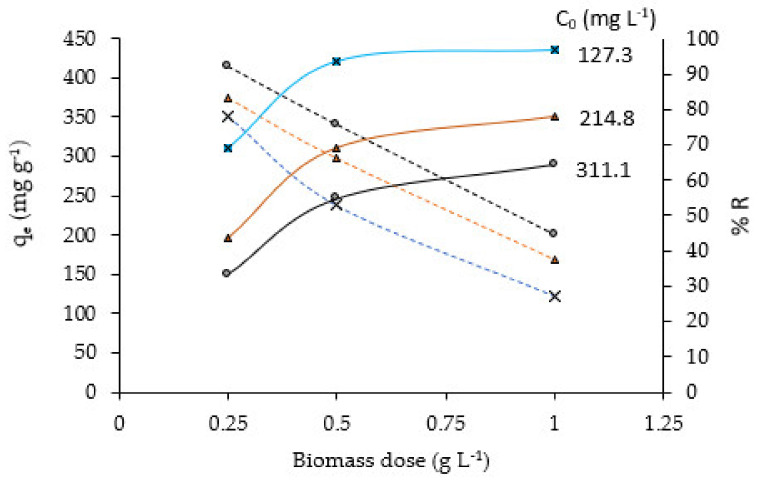
q_e_ (dotted) and %R (continuous) vs. biomass dose plots at different C_0_ concentrations. pH = 4.5, t = 60 min.

**Figure 7 molecules-28-00268-f007:**
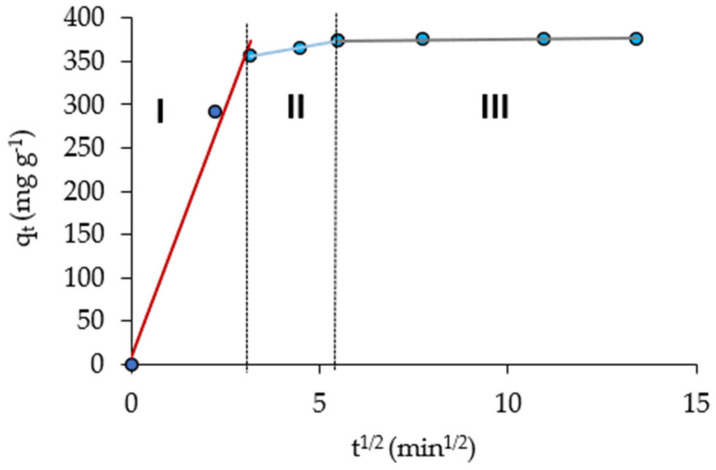
Weber–Morris plots of Pb(II) adsorption on NCM biosorbent.

**Figure 8 molecules-28-00268-f008:**
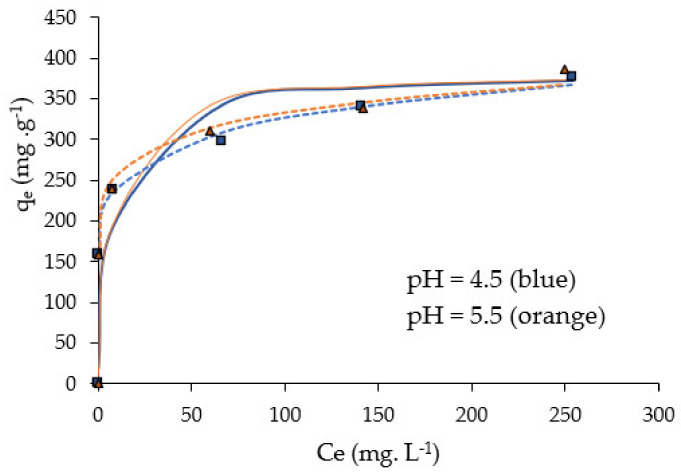
NCM adsorption isotherms fitted to Langmuir (continuous lines) and Freundlich model (dotted lines). For pH 4.5 and pH 5; biosorbent dose = 0.5 g L^−1^, t = 60 min, room temperature.

**Figure 9 molecules-28-00268-f009:**
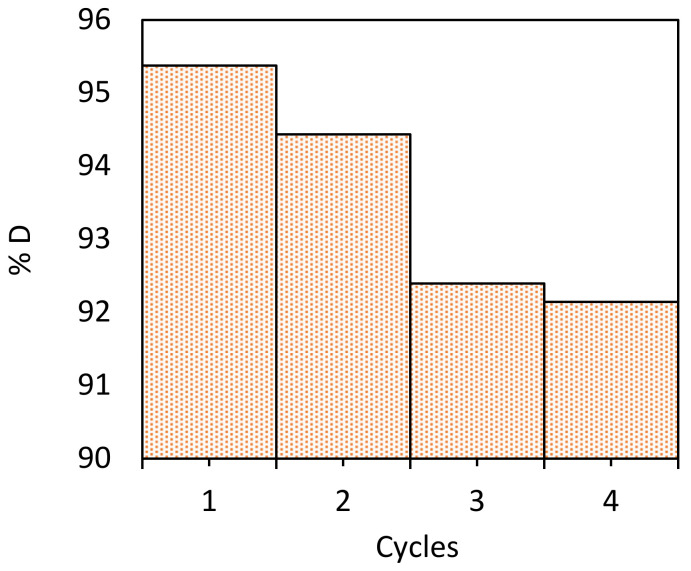
Regeneration efficiency %D of NCM vs. number of Pb(II) adsorption/desorption cycles. Biosorbent dose= 0.5 g L^−1^.

**Figure 10 molecules-28-00268-f010:**
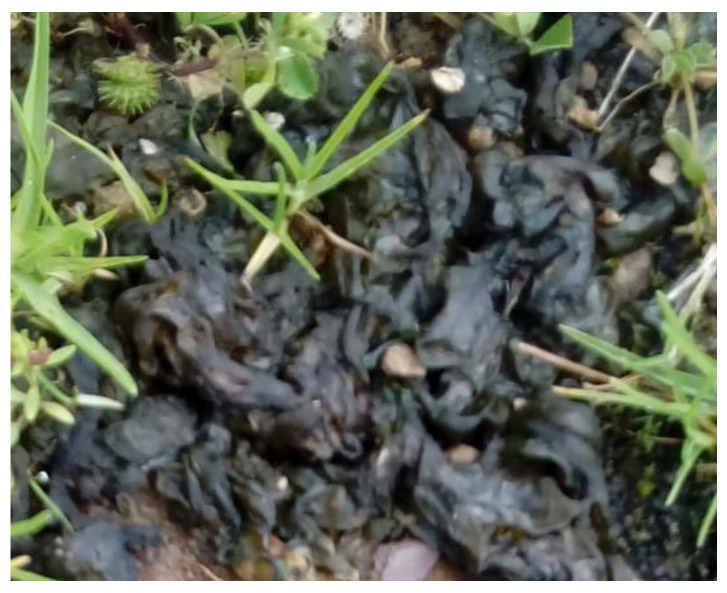
Photograph of gelatinous colony of *Nostoc commune*.

**Table 1 molecules-28-00268-t001:** Concentration of acidic and basic active sites on surface of NC and NCM evaluated by Boehm method.

Biomass	Acidic Sites (mmol g^−1^)	Basic Sites (mmol g^−1^)
Untreated, NC	0.52	0.02
Treated, NCM	0.38	0.11

**Table 2 molecules-28-00268-t002:** Kinetic parameters for the adjustment of experimental data using kinetic models. T = 293 K, C_0_ = 442.42 mg·L^−1^.

Model	Parameters	NCM Biosorbent
*Pseudo* first-order	q_e,cal_ (mg g^−1^) ^(a)^	5.34
	k_1_ (min^−1^)	0.041
	R^2^	0.6
*Pseudo* second-order	q_e,cal_ (mg g^−1^) ^(a)^	384.6
	k_2_ (g mg^−1^⋅min^−1^)	0.0042
	h (mg g^−1^⋅min^−1^)	624.99
	R^2^	1
Elovich	β´ (g mg^−1^)	0.16
	α ×10^6^ (mg g^−1^·min^−1^)	4.27
	R^2^	0.8
Intra-particle diffusion	kid, _I_ (mg g^−1^·min^−1/2^)	115.9
	B_I_	7.25
	R^2^	0.99
	kid, _II_ (mg g^−1^·min^−1/2^)	7.4
	B_II_	332.84
	R^2^	1
	kid, _III_ (mg g^−1^·min^−1/2^)	0.3
	B_III_	372.23
	R^2^	0.81

^(a)^ q_e,cal_: calculated adsorption capacity.

**Table 3 molecules-28-00268-t003:** Adjustment parameters of Pb (II) biosorption isotherms on NCM.

		pH	
Model	Parameters	4.5	5.5
Langmuir	q_max_ (mg g^−1^) ^(a)^	384.6	384.6
	K_L_ (L mg^−1^)	0.12	0.13
	R^2^ χ^2^	0.99 1.6	0.99 7.6
Freundlich	n	7.7	9.1
	K_F_ (mg L^1/n^ g^−1^ mg^−1/n^)	178.5	200.4
	R^2^ χ^2^	0.99 0.6	0.98 1.5
Dubinin–Radushkevich (D–R)	B_DR_ (mol^2^ kJ^−2^)	2 × 10^−8^	7 × 10^−8^
	q_max_ (mg g^−1^) ^(a)^	327.3	309.3
	E (kJ mol^−1^)	500	2673
	R^2^ χ^2^	0.67 34.8	0.67 36.3
Temkin	B (J mol^−1^) K_T_ (Lg^−1^) R^2^ χ^2^	75.3 256.16 0.96 2.27	89.6 217.41 0.93 4.51

^(a)^q_max_: maximum biosorption capacity.

**Table 4 molecules-28-00268-t004:** Comparative table of the Pb(II) biosorption capacity q_e_ of biomasses and agro-industrial wastes.

Biomass	Treatment	q_e,max_ (mg·g^−1^)	Reference
Algae *Nostoc* sp Algae *Oedogonium* sp	Non Non	93.5 145.0	Gupta & Rastogi [9]
Algae *Cystoseira stricta*	NaOH	65	Iddou et al. [19]
Olive Stone	NaOH NaOH	15 ≤25.48	Blázquez et al. [4] Ronda et al. [20]
Maize stover	HNO_3_	27.1	Guyo et al. [46]
Rice bran	NaOH	78.9	Ye and Yu [47]
*Mangifera indica* seed shells	NaOH Carboxyl functionalized	59.25 306.33	Moyo et al. [48]
*Moringa oleifera* tree leaves	NaOH	209.554	Reddy et al. [34]
*Nostoc commune*	NaOH	384.6	This work

**Table 5 molecules-28-00268-t005:** Thermodynamic parameters of Pb(II) biosorption on NCM.

∆*H*^o^ (kJ mol^−1^)	∆*S*^o^ (J mol^−1^ K^−1^)	∆*G*^o^ (kJ mol^−1^)		
		293 K	303K	313K
**−65.1**	−200.8	−6.5	−3.7	−2.5

**Table 6 molecules-28-00268-t006:** Metal concentrations in industrial wastewater (real effluent), before (untreated) and after (treated) Pb(II) removal with NCM.

	Metal Concentration (mg L^−1^)			
	Pb	Ca	K	Na
Untreated wastewater	5.85	125.92	50.35	40.32
Treated wastewater	0.12	45.84	45.29	40.14
removal efficiency, % R	97.9	63.6	10.0	0.4

**Table 7 molecules-28-00268-t007:** Models used for evaluation of Pb(II) biosorption onto NCM.

Model	Equation	Parameters
Kinetic model		
Pseudo-first order	$\log(q_{e}-q_{t})=\log(q_{e})-\frac{k_{1}}{2.303}t$	q_e_: adsorption capacity at equilibrium q_t_: amount of Pb(II) retained per unit mass of biosorbent at time t. k_1_: the first-order kinetic constant k_2_: rate constant adsorption h: initial adsorption rate
Pseudo-second order	$\frac{t}{q_{t}}=\frac{1}{k_{2}q_{e}^{2}}+\frac{t}{q_{e}}$ $h=k_{2}q_{e}^{2}$	
Elovich	$q_{t}=\frac{1}{\beta ´}\ln\left(\propto .\beta ´\right)+\frac{1}{\beta ´}\ln\left(t\right)\;\;$	∝: rate constant *β*: constant related to the covered surface and the activation energy by chemisorption
Intra-particle diffusion	$q_{t}=k_{\mathrm{id}}t^{1/2}+c$	k_id_: intraparticle diffusion rate constant c: constant
**Isotherms**		
Langmuir	$\frac{C_{e}}{q_{e}}=\frac{1}{q_{max}K_{L}}+\frac{C_{e}}{q_{max}}$	C_e_: adsorbate concentration at equilibrium q_max_: Langmuir constant related to the maximum biosorption capacity K_L_: Langmuir constant related to the affinity between sorbent and sorbate
Freundlich	$\ln\left(q_{e}\right)=\ln\left(K_{F}\right)+\frac{1}{n}\ln\left(C_{e}\right)\;$	K_F_: equilibrium constant n: constant related to the affinity between sorbent and sorbate.
Dubinin–Radushkevich (D–R)	$\ln\left(q_{e}\right)=\ln\left(q_{\max}\right)-B_{\mathrm{DR}}\;\varepsilon^{2}$ $\varepsilon =\mathrm{RT}\ln(1+\frac{1}{C_{e}})$	Β_DR_: constant related to adsorption energy ε: Polanyi potential
Temkin	$q_{e}=\frac{\mathrm{RT}}{B}\ln(K_{T})+\frac{\mathrm{RT}}{B}\ln C_{e}$	R: universal gas constant T: temperature $K_{T}$: Temkin’s equilibrium constant B: adsorption energy variation

## Data Availability

Not applicable.

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
