# Peer review of "Efficient Lead Pb(II) Removal with Chemically Modified Nostoc commune Biomass"

_molecules, 2022, doi:10.3390/molecules28010268_

Round 1

Reviewer 1 Report

This work discusses the fabrication of a new biosorbent based on Nostoc commune (NC) cyanobacteria, chemically modified with NaOH (NCM) to remove Pb(II) from aqueous solution. The work is important trend for environmental safety by reducing lead pollution as well as produce valuable cheap biomaterials. The results obtained are promised (qe = 384.6 mg g-1). The overall manuscript is well organized and presented. However, some comments are required before publication such as:

1-      Surfaces area is important parameter for adsorbent materials, please, provide the surface area of the prepared adsorbents and correlate the adsorption capacity with it.

2-      Improve the introduction part by mention the previous and effective adsorbents for Pb removal such as:

https://doi.org/10.1007/s12517-021-06667-6

https://doi.org/10.1016/j.arabjc.2020.01.015

Author Response

Point 1: Surfaces area is important parameter for adsorbent materials, please, provide the surface area of the prepared adsorbents and correlate the adsorption capacity with it.

Response 1: We appreciate the referee's observation, however we believe that surface area data would not more information would provide us with more information for the discussion of our results.

Point 2: Improve the introduction part by mention the previous and effective adsorbents for Pb removal such as:

https://doi.org/10.1007/s12517-021-06667-6

https://doi.org/10.1016/j.arabjc.2020.01.015

Response 2: It has been considered

Reviewer 2 Report

1.       In line number 83, capitalize the genus name ulva lactuca into Ulva lactuca.

2.       Add photographs and microscopic photographs of Nostoc commune and confirms the purity of the culture.

3.       If the culture was isolated from Pampa Cruz in the Junín Region-Peru, how the cyanobacterium was identified as Nostoc commune?

Author Response

Point 1: In line number 83, capitalize the genus name ulva lactuca into Ulva lactuca.

Response 1: It has been corrected

Point 2: Add photographs and microscopic photographs of Nostoc commune and confirms the purity of the culture.

Response 1: The Nostoc commune photograph was added (figure 9)

Point 3: If the culture was isolated from Pampa Cruz in the Junín Region-Peru, how the cyanobacterium was identified as Nostoc commune?.

Response 2: The taxonomic identification of the Nostoc commune was carried out in the Natural History Museum of the Universidad Mayor de San Marcos, Lima-Peru, such as can be seen in the certificate issued by this entity.

Reviewer 3 Report

The paper can be published after minor correction:

1. Authors should show the data for sorption isotherms of Pb(II), data and plots. The models: Languir-Freundlich and Temkin should be additionallly applied for data evaluation

2. The change of sorption with pH should be correlated with the speciation of Pb(II) in the aqueous phase

Author Response

Response to Reviewer 3 Comments

Point 1: 1. Authors should show the data for sorption isotherms of Pb(II), data and plots. The models: Languir-Freundlich and Temkin should be additionallly applied for data evaluation

Response 1: The figure 8 was added and the tenkim model included in the table 3.

Point 2: The change of sorption with pH should be correlated with the speciation of Pb(II) in the aqueous phase.

Response 1: It was considered in figure 5
